# CO-ACTIVATION PATTERNS ALGORITHM: A FORWARD-ONLY DESIGN

## ABSTRACT

Traditional end-to-end neural networks are designed to optimise the predictive accuracy of the final output layer, therefore rendering the network dependent on error backpropagation (BP) for training. Although BP has achieved remarkable success across a wide range of tasks, it has been criticised for its reliance on precise long-range gradient transmission, weight symmetry, and sequential learning constraints. Inspired by co-activation patterns (CAPs) in neuroscience, we propose a learning framework centred on separability of diffrent patterns to circumvent the dependence on BP. In this framework, the network "output" is redefined as a global activation state aggregated across layers, with the backbone regarded as a pattern extractor. Task discrimination is achieved through the evaluation of cosine similarity between CAPs. From an optimisation perspective, each layer updates its parameters using only its own partial derivatives. This removes the reliance on long-range gradient propagation. Simultaneously, global coupling across layers is maintained through fractional normalisation and inter-class competition. In addition, constraints on the co-activation patterns allow task-specific sub-networks to emerge spontaneously. More importantly, this framework readily extends to cross-modal integration and multimodal joint inference, enabling heterogeneous and independent sub-networks to operate in a loosely coupled manner via CAPs, without weight sharing or long-range gradient exchange. Experimental results across multiple datasets demonstrate that the proposed CAPs-based method achieves comparable accuracy to classical BP while significantly accelerating training.

## 1 INTRODUCTION

Neural networks are commonly designed as end-to-end multi-layer mapping structures (Cybenko, 1989), where each layer functions as a sequential processing unit contributing to the generation of optimal predictions for the final output layer. Consequently, learning across all layers is driven entirely by top-down error feedback, giving rise to a backpropagation (BP)-centred learning mechanism. While BP has served as the computational backbone for many successful architectures, researchers in neuroscience have raised several critical concerns regarding its biological plausibility: (i) biological neurons lack the capacity to store or manipulate large-scale weight matrices, thus preventing precise, long-distance and symmetric gradient transmission; (ii) error signals are unlikely to propagate backwards along the same pathways; (iii) error propagation should occur in a parallel, multi-path manner; and (iv) plasticity in brains does not adhere strictly to sequential locking mechanisms (Crick, 1989; Hinton & McClelland, 1987).

In contrast, theories of consciousness, such as Integrated Information Theory (IIT) (Tononi, 2004), originate from a fundamentally different observation: any conscious experience (i.e., subjective awareness) is composed of rich and heterogeneous informational components (Dehaene & Changeux, 2011), which are integrated at the system level into a unified perceptual state. For instance, perceiving a painting is not merely a stepwise recognition of colour and shape, but rather the holistic fusion of these features into a coherent visual representation. Such experiences reflect the cooperative contributions of distributed neuronal populations in forming global perception. This differs markedly from the layer-wise, quasi-Markovian dependency in conventional neural networks, where the final output is typically determined only by the highest-level abstract representation (Baars, 1993; Shanahan, 2006; Chalasani & Principe, 2013).

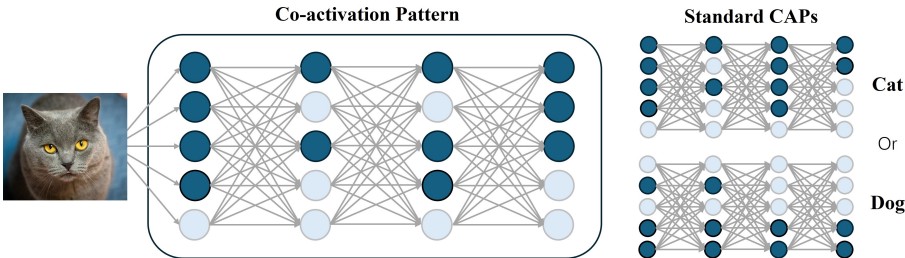

Figure 1: Extraction of Co-activation Patterns: the global activation state serves as the network output, with the backbone limited to forward propagation. Task-specific outputs are derived by analysing similarities between activation modalities.

For a backbone neural network $\mathcal{N}$ comprising $L$ layers, we define perception as the global activation state vector $\mathbf{A}(\mathbf{x})$ induced by an input $\mathbf{x}$ within the network:

$$\mathbf{A}(\mathbf{x}) = \big[\mathbf{a}_1(\mathbf{x}), \mathbf{a}_2(\mathbf{x}), \dots, \mathbf{a}_L(\mathbf{x})\big], \tag{1}$$

where $\mathbf{A}(\mathbf{x}) \in \mathbb{R}^D$, $D = \sum_{l=1}^{L} d_l$ is the total number of neurons in the network, and $\mathbf{a}_l(\mathbf{x})$ denotes the activation vector of layer $l$ under input $\mathbf{x}$. In neuroscience, different neuronal populations exhibit diverse global activation states in response to stimulation. These patterns are commonly referred to as co-activation patterns (CAPs) (Liu et al., 2018). Adopting this terminology, we regard $\mathbf{A}(\mathbf{x})$ as a quantitative representation of the network's internal representational state induced by $\mathbf{x}$.

Within this framework, neural nodes operate under the principle of 'forward correlation and backward independence'. The notion of ' forward correlation' denotes that input signals are processed in a hierarchical order, enabling the network to progressively abstract concepts (Hinton & Salakhutdinov, 2006). In contrast, 'backward independence' implies that the cognitive contribution of each node is independent, such that parameter learning does not rely on interdependencies among nodes. Thus, the backbone network is redefined as a pattern extractor, where the separability of co-activation patterns—rather than direct task-specific outputs—becomes the central design objective. The network propagate unidirectionally forward to generate and preserve cross-layer activation states, while downstream task-specific outputs are obtained via analysis and mapping of the similarity between patterns (Fig. 3). For greater clarity regarding the positioning of our contribution within the existing literature, an extended review of related work is deferred to Appendix A.

We quantify the separability of pattern using strong negative correlation (Liu & Yao, 1999) across several benchmark datasets (Chen et al., 2022) and conduct systematic comparisons. Results demonstrate that, relative to full BP-based training, the proposed pattern-constrained approach exhibits only marginal degradation in optimisation performance, while substantially improving robustness, ability to generalise to a range of tasks and significantly reducing training time. By reducing collinearity among neural units, pattern constraints enhance the decoupling capacity of task-specific sub-networks, offering a promising alternative to error-backpropagation-centric architectures.

## 2 CO-ACTIVATION PATTERNS

Consider a standard feedforward neural network that maps an input $\mathbf{x} \in \mathbb{R}^{d_0}$ to a final-layer representation $\mathbf{a}_L \in \mathbb{R}^{d_L}$. This mapping is realised through the composition of $L$ layers:

$$\mathbf{a}_i = f_i(\mathbf{a}_{i-1}) = \sigma_i\big(W_i \mathbf{a}_{i-1} + \mathbf{b}_i\big), \quad i = 1, \dots, L, \qquad \mathbf{a}_0 = \mathbf{x}, \tag{2}$$

where $\sigma_i$ denotes an element-wise nonlinear function, and $(W_i, \mathbf{b}_i)$ are trainable parameters. In the conventional end-to-end training, the loss function is defined as $\mathcal{L} = \mathcal{L}(\mathbf{a}_L, y)$, depending solely on the relation between the output $\mathbf{a}_L$ and the target $y$. To enable a multi-layer nonlinear structure to approximate complex mappings, informative error signals must be propagated backwards from the

output layer to preceding layers. The error signal $\frac{\partial \mathcal{L}}{\partial \mathbf{a}_i}$ is computed recursively as

$$\frac{\partial \mathcal{L}}{\partial \mathbf{a}_i} = \left( \prod_{k=i+1}^{L} \frac{\partial \mathbf{a}_k}{\partial \mathbf{a}_{k-1}} \right)^{\top} \frac{\partial \mathcal{L}}{\partial \mathbf{a}_L}, \qquad \frac{\partial \mathbf{a}_k}{\partial \mathbf{a}_{k-1}} = \mathrm{Diag}\big(\sigma_k'(W_k \mathbf{a}_{k-1} + \mathbf{b}_k)\big) W_k. \tag{3}$$

This expression involves a product of Jacobian matrices and is therefore prone to vanishing or exploding gradients (Bengio et al., 1994), governed by the spectral properties of these matrices. While differentiability of $\sigma_i$ with a moderate slope constitutes a standard sufficient condition for stable gradient transmission (often complemented in practice by orthogonal or variance-preserving initialisation of $W_i$) (Mishkin & Matas, 2015; Xiao et al., 2018), strong invertibility assumptions on $\sigma_i$ or $W_i$ are not strictly necessary for the correctness of backpropagation. Rather, they are technical assumptions invoked in specific analyses to improve the local conditioning of the Jacobians.

From a system perspective, we replace the explicit output vector with CAPs, and reinterpret the forward pass $\mathbf{a}_0 \to \mathbf{a}_1 \to \cdots \to \mathbf{a}_L$ as a unidirectional mapping from low-level to high-level features:

$$\mathcal{F} : \mathbb{R}^{d_0} \to \prod_{i=1}^{L} \mathbb{R}^{d_i}, \qquad \mathbf{x} \mapsto \mathcal{A}(\mathbf{x}) = (\mathbf{a}_1(\mathbf{x}), \ldots, \mathbf{a}_L(\mathbf{x})), \tag{4}$$

where $\mathbf{a}_i \in \mathbb{R}^{d_i}$ denotes the activation at layer $i$ (the raw input $\mathbf{a}_0 = \mathbf{x}$ is omitted hereafter to avoid notational overhead). The "representational target" of the network is formed collectively by activations across all layers, rather than by a single 'final' vector. Based on $\mathcal{A}(\mathbf{x})$, a general loss functional of contrastive or similarity type can be denoted

$$\mathcal{L}\big(\mathcal{A}(\mathbf{x}), \mathcal{A}(\mathbf{x}^{(j)})\big), \tag{5}$$

where $\mathcal{A}(\mathbf{x})$ denotes the activation set of the current input $\mathbf{x}$, and $\mathcal{A}(\mathbf{x}^{(j)})$ corresponds to that of other samples in the dataset. This viewpoint aligns with recent methods that dispense with an explicit output layer and instead, it discriminates via statistics of the activation patterns. For instance, the Forward-Forward algorithm proposed by Hinton (2022) operates directly in parameter space, learning to classify data by amplifying the difference in activation strengths between positive and negative samples.

To render the above concept more explicit and actionable in classification tasks, we adopt a practical rule that combines unified supervision with gradient decoupling. Specifically, we flatten and concatenate activations across all layers to construct a single global embedding

$$m(\mathbf{x}) = \mathrm{concat}\big(\mathrm{vec}(\mathbf{a}_1(\mathbf{x})), \ldots, \mathrm{vec}(\mathbf{a}_L(\mathbf{x}))\big) \in \mathbb{R}^D, \quad D = \sum_{i=1}^{L} d_i, \tag{6}$$

which provides a common coordinate system in $\mathbb{R}^D$ for comparing all samples. This naturally allows the introduction of trainable class-prototype CAPs, $\mu_c \in \mathbb{R}^D$, for each class $c$ (Algorithm 1), serving as surrogates for full dataset traversal. As an illustrative case, we adopt a cosine similarity with softmax formulation:

$$s_c = \frac{m^{\top} \mu_c}{\|m\|\,\|\mu_c\|}, \qquad p_c = \frac{\exp(s_c/\tau)}{\sum_k \exp(s_k/\tau)}, \qquad \mathcal{L}_{\mathrm{glob}}(m, y) = -\log p_y, \tag{7}$$

where $\tau > 0$ is a temperature parameter, $s_k$ denotes the score with respect to class $k$, and the index $k$ runs over all candidate classes. Denoting $\hat{m} = m/|m|$ and $\hat{\mu}_c = \mu_c/|\mu_c|$, the single-sample gradient becomes

$$\frac{\partial \mathcal{L}_{\mathrm{glob}}}{\partial m} = \sum_c (p_c - \mathbb{1}[c=y])\frac{\partial s_c}{\partial m}, \qquad \frac{\partial s_c}{\partial m} = \frac{\hat{\mu}_c - s_c\,\hat{m}}{\|m\|}. \tag{8}$$

Partitioning $m$ into blocks $m = (m_1, \ldots, m_L)$, where $m_i = \mathrm{vec}(\mathbf{a}_i)$, yields the partial derivative with respect to the $i$-th block:

$$\frac{\partial \mathcal{L}_{\mathrm{glob}}}{\partial m_i} = \frac{1}{\|m\|}\left[ \sum_c (p_c - \mathbb{1}[c=y])\big(\hat{\mu}_{c,i} - s_c\,\hat{m}_i\big) \right], \tag{9}$$

where $\hat{\mu}_{c,i}$ and $\hat{m}_i$ denote the components of $\hat{\mu}_c$ and $\hat{m}$ corresponding to the $i$-th block.

To avoid conceptual ambiguity, we first clarify the definitions of partial and total derivatives. If $(m_1, \ldots, m_L)$ are treated as mutually independent block variables, the above expression provides the partial derivative with respect to the $i$-th block, namely the instantaneous sensitivity of $\mathcal{L}_{\text{glob}}$ to $m_i$ while holding the other blocks fixed. In contrast, for a full gradient backpropagation, $\mathbf{a}_i$ also affects the loss via deeper blocks $\{m_j\}_{j>i}$, in which case, the total derivative with respect to $\mathbf{a}_i$ should be written as

$$\underbrace{\frac{\partial \mathcal{L}_{\text{glob}}}{\partial m_i}}_{\text{direct term (partial derivative)}} + \underbrace{\sum_{j>i} \left(\frac{\partial \mathbf{m}_j}{\partial \mathbf{m}_i}\right)^{\top} \frac{\partial \mathcal{L}_{\text{glob}}}{\partial m_j}}_{\text{indirect term (propagated via deeper layers)}}. \tag{10}$$

The update strategy adopted in this work is equivalent to performing block coordinate descent (or ascent) on $\mathcal{L}_{\text{glob}}(m_1, \ldots, m_L)$. When updating the $i$-th layer, the remaining blocks $\{m_j\}_{j \neq i}$ are treated as constants. Consequently, the update uses the partial derivative $\partial \mathcal{L}_{\text{glob}}/\partial m_i$ while discarding all indirect terms propagated along $\mathbf{m}_i \to \mathbf{m}_j$ $(j > i)$. In implementation, this operation of "treating the other blocks as constants" corresponds precisely to applying a stop-gradient between adjacent layers: the computational graph edges from $\mathbf{a}_i$ to deeper layers are severed, such that the automatic differentiation system returns only the partial derivative. Thus,

$$\frac{\partial \mathcal{L}_{\text{glob}}}{\partial \theta_i} = \left(\frac{\partial \mathcal{L}_{\text{glob}}}{\partial \mathbf{a}_i} \odot \sigma_i'(Z_i)\right) \frac{\partial Z_i}{\partial \theta_i}, \qquad Z_i = W_i \mathbf{a}_{i-1} + \mathbf{b}_i, \qquad \frac{\partial \mathcal{L}_{\text{glob}}}{\partial \mathbf{a}_i} \equiv \frac{\partial \mathcal{L}_{\text{glob}}}{\partial \mathbf{m}_i}, \tag{11}$$

where $\theta_i = (W_i, \mathbf{b}_i)$ denotes the parameters of the $i$-th layer. In other words, starting directly from the similarity objective, updating the $i$-th layer with all other blocks frozen naturally yields an update rule that retains only the direct term. This does not modify the objective function itself, but rather applies a block-coordinate partial-derivative update to the same objective.

It is important to emphasise that, although the indirect terms of end-to-end backpropagation are discarded at the computational-graph level via stop-gradient, the global coupling induced by the discriminative objective still holds. The reason is that the partial derivative $\partial \mathcal{L}_{\text{glob}}/\partial m_i$ explicitly depends on $|m|$, $\hat{m}$, $s_c(m)$, $p_c(m)$, and all class-prototype CAPs $\mu_c$ –quantities jointly determined by the entire activation chain $\{m_j\}_{j=1}^{L}$. Consequently, block updates are modulated by global normalisation and class competition, rather than realised through Jacobian chains propagating "from deeper to shallower" layers. Formally, for all $\forall i \neq j$, $\partial^2 \mathcal{L}_{\text{glob}}/\partial m_i \partial m_j^{\top} \neq 0$, which reflects cross-layer interactions. Put differently, we perform block-coordinate partial-derivative updates on the same global objective $\mathcal{L}_{\text{glob}}$ ; what is discarded are the indirect propagation paths in the graph, not the intrinsic global consistency constraints of the objective.

Nevertheless, ignoring the chained term $\sum_{j>i}(\partial m_j/\partial m_i)^{\top}(\partial \mathcal{L}_{\text{glob}}/\partial m_j)$ means that the update direction no longer coincides with the true steepest descent. When the overall scale of the representation $\|m\|$ fluctuates during training, this bias is further amplified, since the block gradient satisfies

$$\left\|\frac{\partial \mathcal{L}_{\text{glob}}}{\partial m_i}\right\| \propto \frac{1}{\|m\|}. \tag{12}$$

On the one hand, if $\|m\|$ increases, the effective step size of all block parameters is simultaneously reduced, causing learning-rate mismatch and slower convergence. On the other hand, the model may deliberately enlarge the norm ("norm inflation") to reduce gradient magnitude, leading to norm competition and other scale pathologies (Zhang et al., 2018). To mitigate these issues, we incorporate Batch Normalisation (BN) (Ioffe & Szegedy, 2015) in each layer, centring the activations $a_i$ and normalising them by the mini-batch variance, while introducing a learnable scaling factor $\gamma_i$ to adjust block scale. This ensures that $\mathbb{E}|m|^2 \approx \sum_i d_i$, and $\overline{\gamma_i^2}$ is effectively controlled in expectation, thereby stabilising the global step size induced by $1/\|m\|$. Furthermore, the reduction of cross-layer distributional shift can indirectly alleviate directional bias. In small-batch or single-sample settings, Layer Normalisation ba2016layer may serve as a viable alternative.

## 3 TRAINING PROCEDURE

The CAPs-based training procedure is summarised in Algorithm 1. Specifically, the activations from each block are subjected to batch normalisation and pooling, before being concatenated into a global pattern vector. This representation is subsequently aligned with the class template matrix and constrained via a softmax function, after which gradient-based optimisation is employed to update the parameters. Details of the network architecture, dataset descriptions, and implementation details are presented in Appendix B.

---

**Algorithm 1** Training Step of Co-activation Patterns Algorithm

---

**Require:** dataset $\{(x_i, y_i)\}_{i=1}^{B}$, batch size $B$, network $f_\theta$, average pooling $\mathrm{P}(\cdot)$, batch norm $\mathrm{BN}(\cdot)$, class template matrix $T \in \mathbb{R}^{C \times H}$
**Ensure:** updated $\theta$, $\mathrm{BN}$, $T$
  1: **for** each mini-batch $\mathcal{B} \subset \{(x_i, y_i)\}_{i=1}^{N}$ of size $B$ **do**
  2:      initialise empty activation buffer: $global\_state \leftarrow [\,]$
  3:      **for** each block $b$ in $f_\theta$ **do**
  4:          $x_b \leftarrow b.\mathrm{CONV}(x)$
  5:          $x \leftarrow \mathrm{stopgradient}(x_b)$
  6:          $global\_state.\mathrm{append}(\mathrm{flatten}(\mathrm{BN}(\mathrm{P}(x_b))))$
  7:      $h \leftarrow \mathrm{concat}(global\_state)$                   ▷ Global pattern vector $h \in \mathbb{R}^{B \times H}$
  8:      logits $\leftarrow h \cdot T^{\top}$
  9:      $p \leftarrow \mathrm{softmax}(\text{logits})$
 10:      compute the loss: $\mathcal{L} = -\frac{1}{B} \sum_{i \in \mathcal{B}} \log p_{i, y_i}$
 11:      update $\theta$, $\mathrm{BN}$, and $T$ using gradient-based optimisation (e.g. Adam)

---

## 4 RESULTS AND DISCUSSION

This section presents a systematic empirical evaluation of the CAPs-based separability quantification framework in comparison with the conventional end-to-end training approach. The analysis focuses on four aspects: classification accuracy, convergence behaviour, gradient directional consistency, and scale stability. For reference, the standard implementation of "cosine similarity + softmax" under end-to-end backpropagation (formally equivalent to classical cross-entropy) is adopted as the baseline loss. It should be emphasised that, although the two are notationally identical, their algorithmic roles differ significantly. In the CAPs framework, global normalisation and class-prototype CAPs competition induce strong competitive coupling across layers and blocks, leading to marked negative correlation (mutual inhibition) among their contributions. By contrast, end-to-end cross-entropy mainly performs probability modelling and normalisation at the output layer, and its constraints on intermediate layers remain relatively indirect and weak.

### 4.1 THE EFFECT OF NETWORK DEPTH

In terms of accuracy, Table 1 shows that error rates under both paradigms decrease as the backbone depth increases, consistent with the well-established expectation that deeper convolutional networks provide representational benefits. On the Multilayer Perceptron (MLP), CAPs achieves a substantially lower average error than output layer class training using BP, whereas within the Visual Geometry Group (VGG) series no substantial difference is observed. Since MNIST-family tasks are largely saturated, these differences alone are insufficient to determine superiority; more discriminative conclusions require evaluation on more challenging natural image datasets.

In terms of cost and efficiency, Table 1 shows that CAPs, compared with BP, exhibits the following overall characteristics: a slight increase in parameters, reduced memory footprint, markedly faster backpropagation, and forward time that remains largely comparable or shows only minor fluctuations. The increase in parameter count primarily arises from the trainable class-prototypes CAPs $\mu_c \in \mathbb{R}^{D}{}_{c=1}^{C}$: a finite set of prototype vectors replaces exhaustive traversal of the dataset, introducing approximately $D \times C$ additional parameters. In our implementation, features from each layer are globally averaged and unified to 1024 dimensions, so that $D \approx 1024L$, and the extra parameters therefore grow linearly with both depth and number of classes. In the VGG series, the increase

Table 1: Performance comparison of BP and CAPs across varying-depth backbones on MNIST, Fashion-MNIST and Kuzushiji-MNIST considering Error, Parameters, Memory, and Runtime (ms).

| Backbone | Method | MNIST Error | | | Params(M) | Memory(MiB) | Runtime (ms) | |
|---|---|---|---|---|---|---|---|---|
| | | Origin | Fashion | Kuzushiji | | | Forward | Backward |
| 3×3000 MLP | BP | 4.29% | 16.69% | 2.82% | 0.03 | 231 | 0.61 | 1.36 |
| | CAPs | 1.79% | 12.81% | 1.91% | 0.09 | 235 | 0.62 | 0.88 |
| VGG6 | BP | 0.30% | 0.44% | 5.07% | 4.46 | 2092 | 1.28 | 3.44 |
| | CAPs | 0.39% | 0.51% | 4.97% | 4.54 | 1941 | 1.15 | 2.35 |
| VGG8 | BP | 0.29% | 0.40% | 4.50% | 5.24 | 2330 | 2.85 | 6.04 |
| | CAPs | 0.33% | 0.38% | 4.42% | 5.36 | 2181 | 2.82 | 5.05 |
| VGG11 | BP | 0.34% | 0.39% | 4.32% | 7.96 | 2695 | 3.68 | 8.20 |
| | CAPs | 0.33% | 0.40% | 4.29% | 8.14 | 2549 | 4.17 | 5.81 |
| VGG16 | BP | 0.32% | 0.37% | 4.69% | 13.67 | 3158 | 5.44 | 11.57 |
| | CAPs | 0.35% | 0.35% | 5.68% | 13.95 | 3008 | 5.53 | 8.89 |
| VGG19 | BP | 0.34% | 0.35% | 3.80% | 16.83 | 3413 | 6.20 | 13.40 |
| | CAPs | 0.39% | 0.48% | 5.59% | 17.18 | 3257 | 6.61 | 9.71 |

is typically confined to single-digit percentages, whereas for the extremely small-parameter MLP, the relative increase is larger but the absolute scale remains negligible. Training memory usage decreases by about 5–7% across VGG series, reflecting the effect of stop-gradient in avoiding layer-by-layer storage and backpropagation of long Jacobian chains, hence shortening the activation and gradient window that must be retained; convolution-dominated networks benefit more noticeably. With respect to runtime, the backward phase consistently achieves double-digit percentage speed-ups (16–35%), attributable to the elimination of end-to-end chained indirect terms, which allows parallel updates. In the forward phase, the additional computation of the global embedding and cosine similarity with $C$ prototypes has complexity $O(D, C)$, which is of the same order as the linear classifier head in BP. Relative to the convolutional backbone, this extra operator is usually of secondary magnitude, and thus overall forward time shows only minor fluctuations.

## 4.2 The Effect of the Number of Classes

Table 2: Performance comparison of BP and CAPs across Backbones on CIFAR-10 and CIFAR-100 considering Error, Parameters, Memory, and Runtime (ms).

| Backbone | Method | CIFAR-10 | | | | | CIFAR-100 | | | | |
|---|---|---|---|---|---|---|---|---|---|---|---|
| | | Error | Params (M) | Mem (MiB) | Fwd (ms) | Bwd (ms) | Error | Params (M) | Mem (MiB) | Fwd (ms) | Bwd (ms) |
| 3×3000 MLP | BP | 55.66% | 0.03 | 289 | 0.52 | 0.74 | N/A | N/A | N/A | N/A | N/A |
| | CAPs | 48.55% | 0.09 | 294 | 0.50 | 1.12 | N/A | N/A | N/A | N/A | N/A |
| VGG6 | BP | 8.69% | 6.03 | 2212 | 1.49 | 4.27 | 32.86% | 6.86 | 2216 | 1.84 | 4.24 |
| | CAPs | 10.34% | 6.12 | 2012 | 1.65 | 3.14 | 32.88% | 7.68 | 2018 | 1.90 | 3.11 |
| VGG8 | BP | 6.23% | 6.81 | 2517 | 2.76 | 6.03 | 29.80% | 8.01 | 2522 | 2.39 | 8.96 |
| | CAPs | 8.36% | 6.94 | 2319 | 2.87 | 4.74 | 30.57% | 9.24 | 2329 | 2.86 | 5.25 |
| VGG11 | BP | 5.82% | 9.53 | 2981 | 4.45 | 9.49 | 28.31% | 11.28 | 2989 | 4.56 | 9.39 |
| | CAPs | 8.30% | 9.72 | 2787 | 4.85 | 6.96 | 29.98% | 13.13 | 2800 | 4.93 | 6.46 |
| VGG16 | BP | 5.66% | 15.24 | 3573 | 6.60 | 13.65 | 32.31% | 17.92 | 3584 | 7.09 | 14.59 |
| | CAPs | 8.71% | 15.53 | 3369 | 6.89 | 9.82 | 30.88% | 20.78 | 3391 | 8.46 | 12.02 |
| VGG19 | BP | 5.83% | 18.40 | 3899 | 7.88 | 16.35 | 34.14% | 21.63 | 3913 | 7.92 | 16.52 |
| | CAPs | 9.20% | 18.75 | 3685 | 8.07 | 11.53 | 31.53% | 25.11 | 3711 | 8.76 | 12.35 |

We conducted a systematic comparison of BP and CAPs on CIFAR-10 and CIFAR-100 (see Table 2). On CIFAR-10, CAPs clearly outperforms BP on the shallow MLP $3\times3000$. For convolutional backbones in the VGG family, CAPs yields slightly higher error rates than BP, with an average difference of $+2.54$ percentage points across the five depths. On CIFAR-100, however, the opposite trend emerges: while CAPs is comparable to but marginally inferior to BP on VGG6, VGG8, and VGG11, it achieves a consistent advantage on the deeper VGG16 and VGG19, reducing error rates by 1.43 and 2.24 percentage points, respectively.

The observed differences arise from the coupling of global normalisation and class-prototype competition in CAPs. Under a fixed norm budget, class prototypes engage in strongly exclusive competition via the softmax, inducing a systematic negative correlation in discriminative contributions across layers: when one layer gains discriminative strength, the effective contribution of others is correspondingly suppressed. On CIFAR-100 (with more classes and stronger competition), achieving higher posterior quality drives representational capacity towards deeper layers to form sharper margins, while shallow layers converge to shared low-frequency and shape priors, thereby reducing overfitting. Consequently, deeper backbones (VGG16/19) attain a stable advantage. By contrast, on CIFAR-10, capacity is not sufficiently redistributed to the deeper layers, leaving their potential underutilised. As a result, CAPs exhibits mild degradation on convolutional backbones, whilst still conferring benefits on MLPs with weaker shallow representations.

## 4.3 The Effect of Network Width

Table 3: Performance comparison of BP and CAPs across varying-width backbones on CIFAR-100 considering Error, Parameters, Memory, and Runtime (ms).

| Width | Method | VGG11 | | | | | VGG16 | | | | |
|---|---|---|---|---|---|---|---|---|---|---|---|
| | | Error | Params (M) | Mem (MiB) | Fwd (ms) | Bwd (ms) | Error | Params (M) | Mem (MiB) | Fwd (ms) | Bwd (ms) |
| $1\times$ | BP | 28.31% | 11.28 | 2989 | 4.56 | 9.39 | 32.31% | 17.92 | 3584 | 7.09 | 14.59 |
| | CAPs | 30.55% | 13.13 | 2800 | 4.93 | 6.46 | 30.88% | 20.78 | 3391 | 8.46 | 12.02 |
| $2\times$ | BP | 26.23% | 35.07 | 8290 | 13.35 | 23.71 | 29.02% | 58.51 | 9519 | 21.33 | 38.20 |
| | CAPs | 29.32% | 36.91 | 7912 | 13.50 | 15.41 | 29.84% | 61.38 | 9122 | 21.65 | 24.41 |
| $4\times$ | BP | 25.11% | 57.72 | 14493 | 34.00 | 60.23 | 28.59% | 89.79 | 16240 | 54.40 | 95.42 |
| | CAPs | 28.42% | 57.57 | 14103 | 34.14 | 35.90 | 29.00% | 92.66 | 15856 | 54.59 | 57.22 |

The effects of depth and width on CAPs are also differentiated. Taking the VGG11 family as an example, as width increases from the baseline to $2\times$ and $4\times$, the error of BP decreases steadily, whereas CAPs improves more slowly. By contrast, on the VGG16 baseline CAPs actually outperforms BP (a 1.43 percentage point advantage). This suggests that, once depth is sufficient to support semantic specialisation across layers, the global geometric constraint induced by full-layer embeddings and prototype competition becomes effective: shallow layers capture shared priors, while deeper layers carry more selective features, leading to a more balanced allocation under the fixed "norm budget". Simply widening shallow networks intensifies mutual inhibition around the fixed budget, whereas increasing depth is more conducive to shifting representational capacity towards deeper layers. At the same time, as parameter count and activation size grow, the computational and memory cost of BP backpropagation scales nearly linearly, or even super-linearly, while CAPs reduces backpropagation paths and activation storage by truncating cross-layer gradient chains, and its parallelisation advantage scales markedly with model size.

## 4.4 The Effect of Gradient Indirect Terms
The table also compares VGG11-based "CAPs" (retaining only partial derivatives with stop-gradient) and "CAPs-full" (optimising the same global geometric objective but retaining full-chain indirect terms) to assess the trade-offs of discarding indirect gradients. On CIFAR-10 and CIFAR-100, CAPs-full consistently surpasses CAPs in accuracy, and in several cases even outperforms BP. This shows that, in natural image tasks, geometric constraints alone are insufficient to fully correct the directional bias induced by truncated gradient chains; reinstating the indirect terms significantly

Table 4: Performance comparison of BP, CAPs and CAPs-full on CIFAR10 and CIFAR100 with VGG11 backbone considering Error, Parameters, Memory, and Runtime (ms)

| Backbone | Dataset | Method | Error | Params(M) | Memory(MiB) | Forward (ms) | Backward (ms) |
|----------|---------|--------|-------|-----------|-------------|--------------|---------------|
| VGG11 | CIFAR10 | BP | 5.82% | 9.53 | 2981 | 4.45 | 9.49 |
| | | CAPs | 8.30% | 9.72 | 2787 | 4.85 | 6.96 |
| | | CAPs-full | 5.15% | 9.72 | 2995 | 4.83 | 11.85 |
| | CIFAR100 | BP | 28.31% | 11.28 | 2989 | 4.56 | 9.39 |
| | | CAPs | 30.55% | 13.13 | 2800 | 4.93 | 6.46 |
| | | CAPs-full | 24.38% | 13.13 | 3015 | 4.41 | 10.33 |

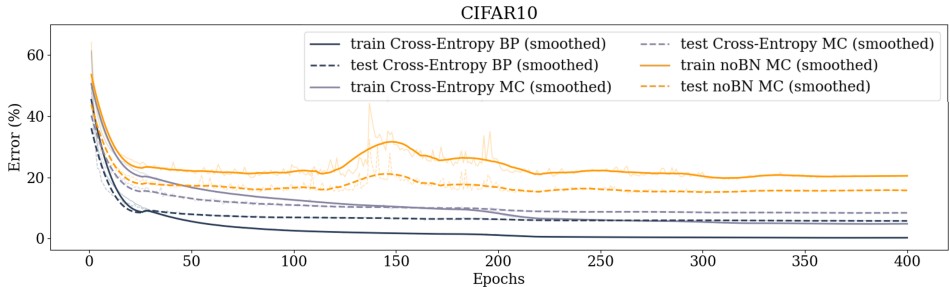

Figure 2: Impact of Batch Normalization ablation (noBN-CAPs) on training and test error dynamics, in comparison with BP and CAPs

reduces this bias. The drawback is increased cost in backpropagation and memory: the backward latency of CAPs-full rises from 6.96 to 11.85 ms (+70%) and from 6.46 to 10.33 ms (+60%) on the two datasets. From an engineering perspective, a practical recommendation is to use CAPs for most of the training to gain substantial advantages in time and memory, and to switch to CAPs-full for short-range fine-tuning in the later convergence stages if ultimate accuracy is required.

### 4.5 THE EFFECT OF BATCH NORMALISATION

Figure 2 shows that batch normalisation (BN) (Ioffe & Szegedy, 2015) is essential for stable training. On CIFAR-10, removing BN leads to a characteristic "scale pathology": the training error remains at a high plateau with pronounced oscillations, while the test error fails to decrease effectively.

The underlying cause lies in the interaction between global normalisation and scale drift. Without normalisation, the norm of the global embedding $m$, obtained by concatenating activations across layers, tends to grow during training, thereby reducing the effective step size to $\eta_{\text{eff}} = \frac{\eta}{\|m\|}$.

If the learning-rate schedule fails to track this drift, training shows "sluggish convergence" and "directional jitter", as shown in the non-BN curves, which exhibit marked kinks and oscillations around epochs 200/300, corresponding to common learning-rate decay points. In contrast, BN recentres and normalises activations using batch statistics, constraining layer-wise scales within a stable range. As a result, the global step size guided by $1/\|m\|$ is stabilised, cross-layer distributional shifts are mitigated, and directional bias is reduced, thereby enabling faster and smoother convergence.

### 4.6 GENERALISATION ABILITY

The test error alone cannot disentangle optimisation dynamics from generalisation ability. To address this, we introduce an analysis of training error to reveal the key dynamics during model learning. As illustrated in Fig. 2, the conventional BP method demonstrates superior optimisation performance, achieving a faster reduction in training error and converging to a lower final value. However, despite its optimisation advantage, BP tends to overfit during compression, over-adapting to training data. In contrast, approaches that emphasise pattern separation display a more balanced behaviour, maintaining resilience against overfitting on both the training and test sets. This consistency suggests that pattern-separation methods inherently possess a stronger inductive bias, favouring solutions with better generalisation capability and thereby mitigating the risk of overfitting.

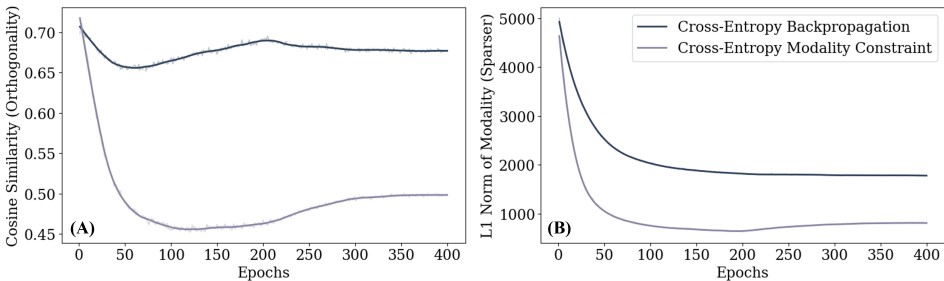

Figure 3: Evolution of Modal Orthogonality and Neuronal Sparsity Across Training Epochs for Various Neural Computation Strategies on CIFAR-10. (A): Cosine Similarity Reflecting Inter-Class CAPs Orthogonality. (B): L1 Norm Reflecting Activation Sparsity in Neural Representations

## 4.7 SPARSITY

Fig.3(A) illustrates the evolution of modal orthogonality during the training of the VGG8 model on the CIFAR-10 dataset, with the aim of assessing the enhancement of inter-class pattern separation, i.e., the independence of task-specific sub-networks. Orthogonality is quantified by computing the average cosine similarity between sample modalities and their corresponding class-mean modalities. The figure is presented as a line plot, where the horizontal axis denotes training epochs and the vertical axis represents the orthogonality metric $O$. Lower values of $O$ indicate stronger inter-class pattern orthogonality. Fig.3(B) visualises the evolution of network sparsity throughout training on CIFAR-10, intended to verify that CAPs under softmax constraints tend to utilise only the minimal number of neuronal nodes required to achieve the computational objective. Similarly, Fig.3(B) is a line plot, with training epochs on the horizontal axis and average L1 norm on the vertical axis. Lower values of $L1_{avg}$ correspond to stronger neuronal sparsity.

As shown in Fig.3(A), standard BP consistently exhibits weak separation of patterns throughout training, lacking clear reasoning pathways. This can be attributed to the absence of explicit pattern constraints, which leads to high correlations between modalities. In contrast, CAPs learning with softmax constraints achieves a marked improvement in orthogonality within the first 200 training epochs. This indicates that, although the primary role of the softmax constraint is to regulate inter-class projection distances, it also effectively facilitates pattern separation during the early stages of training. Correspondingly, Fig.3(B) reveals that BP tends to learn dense weight distributions, with the majority of connections remaining non-zero, thereby increasing redundancy, reducing sparsity, and exacerbating the risk of overfitting. By contrast, pattern-based learning methods demonstrate greater sparsity, which can be ascribed to the independence of their reasoning pathways.

## 5 CONCLUSION

This work introduces a CAPs-centric learning paradigm of "parameter decoupling—representation coupling". In the parameter domain, block-wise updates based on partial derivatives of the global objective enable parallel optimisation while eliminating long-range gradient transmission, thereby reducing training overhead and significantly reducing training time. In the representational domain, alignment through CAPs preserves the necessary global coupling and task consistency. Although the proposed approach shows a modest performance gap compared with standard BP on several benchmarks, its mechanism is more consistent with the local plasticity and parallel propagation observed in biological neural systems. It should be noted that the current implementation relies on engineered operators such as pooling and batch normalisation, which lack direct biological counterparts. In future work, we will pursue two main directions: first, extending this decoupled framework to cross-modal integration and multimodal joint inference, enabling heterogeneous and independent sub-networks to operate in a controllably loosely coupled manner through CAPs without shared weights or long-range gradient exchange; and second, this mechanism makes it possible to introduce new tasks incrementally without damaging existing representations, and also provides a practical computational model for explaining both short-term and long-term memory. Building on this, future work will involve system-level evaluations of its scalability and robustness in lifelong learning and memory consolidation scenarios.

## REPRODUCIBILITY STATEMENT

The underlying principle of our work is simple and readily reproducible. The required implementation steps are presented in Algorithm 1, and all source code is provided in the supplementary material.

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

## A  RELATED WORK

In recent years, numerous training algorithms have emerged that seek to avoid the traditional back-propagation (BP) mechanism. Biologically inspired approaches, such as target propagation (Lee et al., 2015; Bartunov et al., 2018) and feedback alignment (Lillicrap et al., 2014; Nøkland, 2016), utilise auxiliary networks to propagate optimal activations or error signals directly backward, thereby circumventing BP. Decoupled Neural Interfaces (DNI) (Jaderberg et al., 2017) similarly rely on auxiliary networks, but instead generate synthetic gradients. Other techniques, such as the Alternating Direction Method of Multipliers (ADMM), decompose the end-to-end optimisation process into smaller sub-problems using auxiliary variables (Taylor et al., 2016). Decoupled Parallel Backpropagation (Huo et al., 2018b) and Feature Replay (Huo et al., 2018a) use previously computed gradients instead of current ones to update parameters, providing theoretical guarantees of convergence and enabling parallel training of network modules. Collectively, these methods emulate, to some extent, the biological principle of local synaptic plasticity, where neurons adjust synaptic strengths based solely on local input-output relationships. However, these approaches largely remain extensions of end-to-end training paradigms, with optimisation objectives still predominantly defined by errors computed at the output layers or their variants. Crucially, they lack explicit optimisation directly targeting the network's global activation states.

The FF algorithm (Hinton, 2022), similar to our proposed method, explicitly targets the network's activation states. FF entirely abandons BP, instead employing two forward passes: one with genuine data as positive samples and another with artificially generated or differently classed inputs serving as negative samples. Each layer is trained independently by a local objective function, which adjusts weights to increase the activation magnitude for positive samples relative to negative samples, thereby enabling class discrimination. Despite its advantages, FF has two main limitations: firstly, it oversimplifies the evaluation of global network activations by merely summing neuronal activities, failing to exploit the rich, high-dimensional nature of these activation patterns; secondly, its distributed layer-wise optimisation lacks a convincing rationale at earlier layers, as shallow-layer features are forced into a potentially unnatural separation between positive and negative sample activations.

Meanwhile, numerous studies have focused on designing more discriminative loss functions to enhance class separability. These contrastive learning approaches establish a metric space tailored for specific tasks, encouraging representations of samples from the same class to cluster together, while pushing apart those from different classes. Some methods, such as Triplet Loss (Dong & Shen, 2018) and N-Pair Loss (Sohn, 2016; Jaiswal et al., 2020), explicitly measure relative distances between sample pairs. Others, like those proposed by Rippel et al. (2015) and Qi & Su (2017), adjust the representational distances between individual samples and learned class centres. Nevertheless, most of these contrastive losses operate solely on the final layers of the network, emphasising local sample pairs or local statistical differences rather than modelling and controlling the global activation patterns throughout the entire network. Furthermore, all these approaches fundamentally depend on BP for parameter updates. Consequently, individual layers receive fragmented gradient signals originating from the output loss rather than receiving a cohesive and globally informed feedback that directly reflects the overall activation state of the network.

## B  EXPERIMENTAL SETUP

### B.1  NETWORK ARCHITECTURE

In this study, we adopted a multilayer perceptron (MLP) (Krizhevsky et al., 2009) together with a series of VGG-style convolutional networks of varying depths and widths (Simonyan & Zisserman, 2014) as baseline architectures. The $3 \times 3000$ MLP represents a fully-connected structure characterised by short forward paths and relatively low computational and memory overheads, albeit with limited representational capacity. The VGG family adheres to the classical design paradigm: stacking homogeneous $3 \times 3$ convolutional layers (stride of 1, same padding) as the basic building blocks, interleaved with $2 \times 2$ max-pooling layers ($M$) to achieve spatial downsampling. To systematically examine the impact of network depth, we selected five configurations—VGG6, VGG8, VGG11, VGG16, and VGG19. Furthermore, to evaluate the role of width expansion in enhancing representational power, we constructed channel-augmented variants (e.g., VGG11-2x and VGG16-

4x), in which the number of channels is increased by factors of two or four. In all cases, channels were progressively doubled in a stage-wise manner, ranging from 128 channels in the shallow layers up to 1024 channels in the widest models. It is noteworthy that none of these architectures incorporated residual connections or attention mechanisms, thereby ensuring a simplified and controlled setting in which the influence of depth and width on the proposed learning framework could be more transparently assessed.

## B.2 DATASET

We conducted experiments using the PyTorch framework on five benchmark datasets: MNIST (Le-Cun et al., 1998), Fashion-MNIST (Xiao et al., 2017), Kuzushiji-MNIST (Clanuwat et al., 2018), CIFAR-10, and CIFAR-100 (Krizhevsky et al., 2009). MNIST comprises 70,000 handwritten digit images of size $28 \times 28$ pixels, with 60,000 images allocated for training and 10,000 for testing, and is designed for digit recognition. Fashion-MNIST, serving as a more challenging alternative, mirrors the structure of MNIST while consisting of 70,000 images of fashion products. Kuzushiji-MNIST contains cursive Japanese characters and follows the same scale and format as MNIST. Both CIFAR-10 and CIFAR-100 are constructed for image classification tasks, each consisting of 60,000 colour images of size $32 \times 32$ pixels. CIFAR-10 is divided into 10 distinct classes, whereas CIFAR-100 comprises 100 classes further grouped into 20 superclasses.

## B.3 IMPLEMENTATION DETAILS

For experiments on the MNIST family of datasets and CIFAR-10, we employed a $3 \times 3000$ MLP together with VGG variants as baselines to evaluate the performance of different methods; for CIFAR-100, only VGG models were considered as baselines. All experiments were conducted on a single NVIDIA RTX 5090 GPU without data augmentation. The Adam optimiser (Kingma & Ba, 2014) was used with a batch size of 128 and an initial learning rate of $5 \times 10^{-4}$. The number of training epochs was set to 100 for MNIST and Kuzushiji-MNIST, 200 for Fashion-MNIST, and 400 for both CIFAR-10 and CIFAR-100.

A dropout rate of 0.2 was consistently applied to both MLP and VGG models across all datasets. To balance the contribution of individual layers to the CAPs, average pooling was applied to the features of each block, with kernel sizes adjusted as required to ensure that the resulting feature dimension was fixed at 1024 across all datasets.

