# OpenReview forum: "Co-activation Patterns Algorithm: A Forward-only Design"
_ICLR.cc/2026/Conference — ICLR 2026 Conference Withdrawn Submission_

### Official Review · Reviewer_ineg · 2025-10-20

**Soundness:** 3
**Presentation:** 3
**Contribution:** 3
**Rating:** 4
**Confidence:** 3

**Summary:**

The paper reframes a network’s “output” as a concatenated global activation state across layers, dubbed a co-activation pattern (CAP). Class discrimination is done by cosine-similarity to learned class-prototype CAPs. Training performs block-coordinate updates: for each layer, gradients use only the layer’s partial derivative while severing edges to deeper layers (stop-gradient), thereby avoiding long backward chains yet preserving global coupling via the shared similarity objective and normalization. Empirically, CAPs matches or trails BP on CIFAR-10 with VGG backbones but improves on deeper models for CIFAR-100; it provides consistent backward-time speedups and modest memory savings, and requires BatchNorm for stability. A “CAPs-full” variant that restores indirect terms closes the accuracy gap at a higher cost.

**Strengths:**

1. Treating the global, multi-layer activation as the decision object is elegant and well-motivated by CAPs in neuroscience.

2. Depth/width/class-count studies and a BN ablation illuminate when CAPs help and why stability hinges on normalization.

**Weaknesses:**

1. All experiments are conducted on small datasets(MNIST and CIFAR), not large-scale ones. The reviewer is curious about the performance on large-scale datasets such as ImageNet.

2. The method only tests on small models such as VGG and a few-layer MLPs. The reviewer is wondering how CAPs would perform on large-scale backbones such as ResNet variants (ResNet family or ResNext, etc.) and Transformer variants (ViT or Swin).

3. Claims about “norm competition” and global coupling are intuitive but lack convergence/error analyses for block-coordinate updates under BN/LN.

4. “Forward-only” phrasing may be misleading, as gradients still flow locally within each block.

**Questions:**

Please see the weakness above.

---

### Official Review · Reviewer_KJPt · 2025-10-21

**Soundness:** 1
**Presentation:** 2
**Contribution:** 1
**Rating:** 2
**Confidence:** 4

**Summary:**

Backpropagation has enabled remarkable advances in training neural networks, achieving outstanding performance across a wide range of domains. Nevertheless, it has often been criticized for its lack of biological plausibility. This paper introduces the Co-Activation Patterns (CAP) algorithm, a local learning method designed to improve both class separability and biological interpretability. The core idea is to store prototypical class activation patterns (CAPs) and, for each training sample, to compare its feed-forward activations against all stored class prototypes. The similarity scores, computed through a softmax-based matching process, are then used to guide weight updates without relying on global error backpropagation. Empirical results indicate that while backpropagation still achieves higher overall accuracy, the proposed CAP algorithm demonstrates robust generalization and exhibits greater resilience to overfitting, supporting its potential as a biologically inspired alternative learning framework.

**Strengths:**

The paper proposes a local learning technique for neural network training based on class feature matching, contributing a fresh conceptual direction in the study of biologically inspired learning.

 It conducts comprehensive evaluations across multiple dimensions—such as network depth, width, training dynamics, and generalizability—providing a thorough empirical analysis.

 The paper is well-written and accessible, presenting the main ideas clearly and making a complex concept easy to follow.

**Weaknesses:**

The proposed method shows no clear advantage over backpropagation in either accuracy or efficiency. As shown in Figure 2, it consistently underperforms by several percentage points on CIFAR-10 for both training and testing errors. In terms of efficiency, aside from a minor reduction in latency due to asynchronous weight updates, the method offers no substantial improvement in memory or computational cost.

 The explanation of the core component, Class Template, is insufficient. Since comparing each feature with the class template is central to the learning process, scalability becomes a concern: as the number of class templates increases, the computational efficiency would likely degrade sharply. This raises doubts about applicability to domains without well-defined classes, such as language modeling. Moreover, the paper does not clearly describe how class templates are constructed or the computational overhead involved.

 The work lacks comparisons with prior local learning frameworks, such as Hebbian learning[2], equilibrium propagation [1], or the method by [3]. The related works section briefly mentions only the Forward-Forward algorithm without offering deeper conceptual or empirical comparisons to these baselines.

 Although the paper initially aims for biological plausibility, the proposed mechanism arguably diverges from biologically realistic processes. The method relies on explicit similarity computations between current activations and all stored class templates, yet there is no known biological system that categorizes and stores exhaustive feature representations for all classes in such a manner.

**Questions:**

1. The paper emphasizes backward independence, but doesn’t the inclusion of batch (or layer) normalization and CAP matching introduce inter-node dependencies? Normalization is inherently a group operation across nodes, and CAP matching depends on previous activations. Could the authors clarify how this remains locally independent?
2. Is the training stable when weight updates are not explicitly aligned with local gradients? Some clarification or empirical evidence of convergence stability would be valuable.
3. If the class template matrix stores all activation patterns for each class, its size should scale linearly with the number of classes. For large datasets such as ImageNet, this could require several thousand templates—potentially increasing the VRAM requirements by orders of magnitude compared to standard backpropagation. Is this reasoning correct? Are there mechanisms to mitigate such scaling issues?
4. There exist numerous local learning rules, including [1-4]. Why were none of these methods empirically evaluated as baselines?
5. Finally, regarding biological plausibility: is there any biological mechanism that directly corresponds to your proposed learning rule, beyond fragmentary analogies? Given that your method involves storing all class prototype features, it would strengthen the claim of biological plausibility to provide concrete evidence or theoretical justification for such a mechanism.

[1] Scellier, B., & Bengio, Y. (2017). Equilibrium propagation: Bridging the gap between energy-based models and backpropagation. Frontiers in computational neuroscience, 11, 24.

[2] Gerstner, W. (2011). Hebbian learning and plasticity. From neuron to cognition via computational neuroscience, 0-25.

[3] Nøkland, A., & Eidnes, L. H. (2019, May). Training neural networks with local error signals. In International conference on machine learning (pp. 4839-4850). PMLR.

[4] Meulemans, A., Zucchet, N., Kobayashi, S., Von Oswald, J., & Sacramento, J. (2022). The least-control principle for local learning at equilibrium. Advances in Neural Information Processing Systems, 35, 33603-33617.

---

### Official Review · Reviewer_NoH6 · 2025-10-30

**Soundness:** 2
**Presentation:** 3
**Contribution:** 2
**Rating:** 4
**Confidence:** 4

**Summary:**

SUMMARY
The article proposes a forward-only alternative to the backpropagation algorithm for training neural networks. It is inspired by the concept of co-activation patterns (CAPs) in neuroscience, which refer to sets of neurons that become active during specific tasks or in response to certain classes. The proposed algorithm consists of the following main steps:

1.Inference Phase: All hidden activations are stacked into a single large vector, from which the output is generated via a linear transformation followed by a softmax operation.

2. Learning Phase: Only the component of the loss gradient that directly depends on the hidden layer activations is used for learning, while the backpropagated component is ignored.

In essence, connecting each hidden layer directly to the output allows output errors to be projected straight to intermediate layers, while the propagating component is disabled through a stop-gradient operation.

The authors compare their algorithm with backpropagation on the MNIST, CIFAR-10, and CIFAR-100 datasets in terms of classification accuracy, memory usage, and execution time.

**Strengths:**

STRENGTHS

- The article addresses the important problem of developing alternatives to the backpropagation (BP) algorithm. It proposes directly connecting intermediate hidden layers to the output, enabling direct output error feedback to these layers.

- The proposed approach offers some memory savings and a reduction in training update time.

**Weaknesses:**

WEAKNESSES

- The proposed BP alternative seems to aim at two potential goals: (i) providing a biologically plausible substitute for BP, and/or (ii) offering a machine learning algorithm with better accuracy or implementation efficiency than BP. Regarding (i), the assumed architecture—direct projections from all hidden layers to the output—does not appear to be supported by biological evidence. Regarding (ii), the performance is generally inferior to BP, and the reported gains in memory and execution time do not seem substantial.

- The BP-MLP results for the MNIST and CIFAR datasets appear worse than expected. Please refer to Tables 1 and 2 in:

Clark D, Abbott LF, Chung S. Credit assignment through broadcasting a global error vector. Advances in Neural Information Processing Systems, 2021 Dec 6; 34:10053–10066.

In that work, the classification accuracy for MLPs (fully connected networks) trained with BP is higher than what is reported in the article, and it also exceeds the performance of the proposed CAP approach—even when smaller network architectures are used (1024–512 hidden layers for MNIST, and 1024–512–512 for CIFAR-10).

**Questions:**

QUESTIONS

- How well do the CAP gradients with and without the stop-gradient operation align (e.g., in terms of cosine similarity)? What is the quantitative impact of ignoring the propagating term?

- Is there any evidence from functional connectivity studies supporting the biological plausibility of the proposed approach?

- Were the BP experiments conducted with extensive hyperparameter tuning? (Please refer to the reference listed above.)

- How is the performance gap for larger scale inputs?

---

### Official Review · Reviewer_fdU9 · 2025-11-01

**Soundness:** 3
**Presentation:** 3
**Contribution:** 2
**Rating:** 4
**Confidence:** 4

**Summary:**

This paper proposes the Co-activation Patterns (CAPs) algorithm, a forward-only alternative to backpropagation inspired by co-activation patterns in the brain. Instead of propagating errors sequentially through layers as in backpropagation, the model aggregates activations from all layers into a single global representation m(x), and compares it against global trainable prototypes mu_c, via cosine similarity. Each layer is then updated using its own partial derivative, with no long-range gradient flow.

**Strengths:**

The paper is well written on the whole, with a clear mathematical exposition and connection to other learning algorithms such as forward-only and standard backpropagation.

The learning rule is well analysed, including systematic assessment of the effect of network depth and width; indirect gradients, and batch normalisation.

Appendix A contains valuable contextualisation of the approach, though moving some of this material into the main text would help situate the algorithm more clearly in the landscape of alternatives and related work.

**Weaknesses:**

The algorithm relies on learning of global class prototypes via gradient descent, but this is not motivated biologically. It is unclear how these prototypes could be updated or represented across biological neural circuits, limiting the strength of the biological analogy.

The experiments are confined to image datasets with either mlp or convolutional networks. Given the biological inspiration, it is disappointing that CAPs is not extended and applied to Recurrent Neural Networks. Similarly, given the modern dominance of transformers in AI, the absence of transformer-based experiments is a missed opportunity.

Previous work has shown that neuroscience inspired alternatives backpropagation do not scale to harder datasets, namely imagenet (Bartunov 2021, Assessing the Scalability of Biologically-Motivated Deep Learning Algorithms and Architectures). Since CAPs already show slightly worse error rates on CIFAR-10, it is important that CAP is tested on these harder datasets. With modern GPUs and efficient libraries like FFCV such tests are now feasible. At minimum, evaluation on an intermediate benchmark like Tiny ImageNet would strengthen the paper.

Finally, I think the paper could do a better job of more clearly explaining how this work builds on and improves inadequacies of related algorithms.

The reported MNIST-family benchmarks also appear suboptimal. For original and fashion MNIST BP error rates are worse than what should be expected (~< 2%, ~10%). In contrast, K-MNIST is unusually strong, and BP K-MNIST is better than BP MNIST, indicating tuning inconsistencies.

**Questions:**

1. Given the importance of mu_c, this approach seems limited to straightforward classification tasks. Could the authors comment on this?
2. How do the authors interpret mu_c and its optimisation in biological terms?
3. Have the authors considered testing and extending CAP to RNNs or to transformer-based models?
4. Do the authors have evidence or intuition that CAP scales to harder datasets?

---

### Note · Authors · 2026-01-16

I have read and agree with the venue's withdrawal policy on behalf of myself and my co-authors.